# Protective and Vulnerability Factors of Municipal Workers’ Mental Health: A Cross-Sectional Study

**DOI:** 10.3390/ijerph192114256

**Published:** 2022-10-31

**Authors:** Francisco Sampaio, Joana Coelho, Patrícia Gonçalves, Carlos Sequeira

**Affiliations:** 1Higher School of Health Fernando Pessoa, Rua Delfim Maia 334, 4200-253 Porto, Portugal; 2CINTESIS@RISE, Nursing School of Porto (ESEP), Rua Dr. Plácido da Costa, 4200-450 Porto, Portugal; 3Northern School of Health of the Portuguese Red Cross, Rua da Cruz Vermelha Cidacos-Apartado 1002, 3720-126 Oliveira de Azeméis, Portugal; 4Institute of Health Sciences, Universidade Católica Portuguesa, Rua de Diogo Botelho 1327, 4169-005 Porto, Portugal; 5Nursing School of Porto, Rua Dr. António Bernardino de Almeida, 830, 844, 856, 4200-072 Porto, Portugal

**Keywords:** protective factors, occupational groups, mental health, cross-sectional studies

## Abstract

Work is fundamental to an individual’s mental health; however, an unfavourable work environment can lead to mental health problems. Despite existing studies addressing workers’ mental health, it is essential to understand the reality of specific contexts to design effective tailored interventions. Thus, this study aimed to examine the influence of potential protective and vulnerability factors on municipal workers’ depressive symptoms, anxiety and stress levels, and burnout. A cross-sectional study was conducted with data collection performed between July and December 2021 using online self-report measures. The sample comprised 115 municipal workers. The findings revealed that psychological vulnerability is a significant vulnerability factor for the presence of mental health symptoms. In addition, job satisfaction was found to be a significant protective factor for depressive symptoms, anxiety, and burnout of the municipal workers. The results of this study enhance the understanding of factors that influence worker mental health, which may facilitate the proper planning of specific interventions to promote mental health in the workplace.

## 1. Introduction

Mental health is a fundamental component of an individual’s well-being. Moreover, mental disorders are currently one of the most critical public health challenges. According to a study carried out by the Global Burden of Diseases, Injuries, and Risk Factors [1], which evaluated the incidence, prevalence, and years lived with disability for 354 clinical conditions, in 195 countries and territories, from 1990 to 2017, the burden associated with mental disorders is significant. Mental disorders have consistently accounted for more than 14% of years lived with disability for nearly three decades and have a prevalence of over 10% in all regions included in the study [1].

Work is fundamental to individuals’ mental health; however, an unfavourable work environment can lead to physical and mental health problems [2]. Workers’ health, safety, and well-being are of increasing importance worldwide [3]. Mental health problems are common in the working population and represent a growing concern internationally, with potential impact on workers, organizations, health at the workplace, labour markets, and social policies [4]. The relationship between mental illness and the workplace environment is complex and multifaceted. Mental health problems have a negative impact on labour productivity, and adverse work environments are associated with a higher prevalence of mental health problems [5].

Depression and anxiety disorders are the most prevalent mental health disorders worldwide [6], and it is crucial to determine its associated risk factors. According to Nikunlaakso et al., work-related stress may increase the risk of depression, anxiety, burnout, and sleep disorders [7].

A significant body of evidence addresses risk factors associated with the workplace that can negatively affect the physical and mental health of individuals. Furthermore, the workplace is a place that potentiates the development of physical and mental problems but also offers a considerable opportunity for the introduction and development of preventive measures for these problems [8]. Therefore, implementing interventions aimed at promoting mental health at the workplace is a compelling response to the constant challenges in this area [7,8,9,10,11].

According to the World Health Organization [2], inadequate health and safety policies, poor communication and management practices, limited worker participation in decision-making, low levels of worker support, inflexible work schedules, and unclear organisational tasks or objectives are among the risk factors associated with the workplace likely to negatively affect mental health. A healthy workplace is one in which all workers are involved in the process of continuous improvement, intending to protect and promote the health, safety, and well-being of all elements and the sustainability of the workplace [3].

Assessing workers’ mental health indicators is the first step when considering an intervention at this level. Despite existing studies addressing the workers’ mental health, it is essential to understand the reality of specific contexts to design effective tailored interventions. In Portugal, there are several studies on workers’ mental health, but mostly using samples comprised of healthcare workers. Thus, we did not find in the literature any study on municipal workers’ mental health, which is an important gap because, in Portugal, there are 138,258 municipal workers. This study intended to bridge this gap by producing information that can better characterise the national context and potentially be relevant to establish comparisons with international data on this field. This research results from a partnership between the Portuguese Society of Mental Health Nursing and the Center for Health Technology and Services Research (CINTESIS@RISE).

In this study, we refer to mental health status as a concept that involves depressive symptoms, anxiety, stress, and burnout. According to the American Psychiatric Association [12], depressive symptoms are viewed as depressed mood, markedly diminished interest or pleasure in several activities for most of the day, significant weight loss—when not dieting—or weight gain, decreased or increased appetite, fatigue, loss of energy nearly every day, among others. Anxiety is defined as the anticipation of future threats [12]. On the other hand, according to Hans Selye, cited in Fink (2017), *stress is a non-specific response of the body to any demand* [13] (p. 4). Finally, according to Maslach and Leiter [14], burnout is a syndrome that emerges as a response to chronic interpersonal job stressors. This response is composed of three key dimensions: exhaustion, feelings of cynicism and detachment from the job, and a sense of lack of accomplishment and ineffectiveness.

In line with this view, this study sought to answer the following research questions:Do resilience, mental health literacy, job satisfaction, psychological vulnerability, and the number of working hours per week influence municipal workers’ depressive symptoms?Do resilience, mental health literacy, job satisfaction, psychological vulnerability, and the number of working hours per week influence municipal workers’ anxiety levels?Do resilience, mental health literacy, job satisfaction, psychological vulnerability, and the number of working hours per week influence municipal workers’ stress levels?Do resilience, mental health literacy, job satisfaction, psychological vulnerability, and the number of working hours per week influence municipal workers’ burnout?

According to Rutter [15], resilience refers to the ability to overcome stress and/or adversity. Studies suggest that resilience has protective effects on the mental status of individuals facing adversity [16,17].

Mental health literacy has been defined by Jorm et al. [18] as knowledge and beliefs about mental health disorders that allow the individual to recognise, manage, or prevent them. This concept has evolved to integrate components beyond the knowledge regarding mental health disorders, such as understanding how to obtain and maintain good mental health, decreasing stigma related to mental disorders, and enhancing help-seeking efficacy [19]. Bearing in mind that mental health literacy may enhance positive attitudes around mental health [20], it is crucial to investigate if it may have protective effects on mental health status.

Job satisfaction refers to how pleased, satisfied, or comfortable the individual is about his/her job [21]. Studies suggest that job satisfaction is an important factor influencing workers’ health [22,23,24].

Psychological vulnerability is a *pattern of cognitive beliefs reflecting a dependence on achievement or external sources of affirmation for one’s sense of self-worth* [25] (p. 120), and it makes people less protected when facing negative life experiences [25,26].

Several studies have reported the relationship between the number of work hours and workers’ mental health status. Working for long hours has been associated with poorer workers’ mental health [27,28].

## 2. Materials and Methods

### 2.1. Study Design

A cross-sectional study with data collected using online self-report measures was conducted to answer the abovementioned research questions. The research followed the STrengthening the Reporting of OBservational studies in Epidemiology (STROBE) guidelines, a checklist of items to be included in articles reporting observational research [29].

### 2.2. Participants and Setting

The study participants were municipal workers from the Municipality of Felgueiras (a city in the Northern Region of Portugal, composed of 20 parishes with more than 50,000 inhabitants). The Municipality of Felgueiras was granted authorisation by the Portuguese Society of Mental Health Nursing (Portuguese scientific society) to assess its municipal workers’ mental health. Therefore, this study has emerged from the partnership established between both organizations. The inclusion criteria were: (a) to be a municipal worker at the Municipality of Felgueiras; (b) to be 18 years of age or older; and (c) to be able to read and understand European Portuguese. 

### 2.3. Data Collection

The data collection tool was developed by the research team members using Google Forms. Then, it was sent to the Municipality of Felgueiras, which, in turn, resent it by email to all the 340 municipal workers that met the inclusion criteria. The data collection tool was sent by the employer, but only research team members had access to the database. All participants were provided information about the study and reassured about anonymity, to avoid the Hawthorne effect. Municipal workers could include office staff, such as secretaries and other similar assistants, sanitation workers, and planning and zoning staff, such as planners and engineers. Data collection was carried out over six months, from July to December 2021. 

### 2.4. Measures

Our study aimed to assess the protective and vulnerability factors of municipal workers’ mental health. Thus, firstly, to characterise the sample, some demographic and clinical data were collected, such as sex, age, marital status, educational attainment, years at the current job, having a diagnosed mental disorder, and the number of working hours per week.

Mental health status was assessed by measuring the following variables: depressive symptoms, anxiety, stress, and burnout. Depressive symptoms, anxiety, and stress were assessed using the European Portuguese version of the Depression, Anxiety and Stress Scale—21 items (DASS-21) [30], a set of three self-report scales designed to measure the emotional states of depression, anxiety, and stress. Each of the three DASS-21 scales contains seven items, divided into similar content subscales. Scores for depression, anxiety and stress are calculated by summing the scores for the relevant items. Higher scores mean higher levels of depression, anxiety and/or stress. The European Portuguese version of the DASS-21 presents an acceptable internal consistency (Cronbach’s alpha) in all its subscales (depression = 0.60; anxiety = 0.50; stress = 0.50). Burnout was assessed by the European Portuguese version of the Oldenburg Burnout Inventory (OLBI) [31]. It is a self-report five-point rating scale (1 = Strongly disagree; to 5 = Strongly agree) with seven and eight questions within each of the two dimensions, disengagement and exhaustion, respectively. The score is calculated by summing all items’ scores; then, the value is divided by the total number of items on the scale. The higher the score, the greater the level of burnout. The European Portuguese version of the OLBI presents a very good internal consistency (Cronbach’s alpha) (0.93 for the total scale).

To assess the potential protective factors, the following variables were measured: resilience, mental health literacy, and job satisfaction. Resilience was assessed by the European Portuguese version of the Connor–Davidson Resilience Scale (CD-RISC-10) [32]. It is a unidimensional self-report scale consisting of 10 items measuring resilience. Respondents rate items on a 5-point Likert scale ranging from 0 (not true at all) to 4 (true nearly all the time). Total scores are calculated by summing all 10 items. Higher scores indicate higher resilience. The exploratory and confirmatory analysis revealed one dimension and good psychometric properties in a sample of Portuguese individuals. Mental health literacy was assessed by the European Portuguese version of the Mental Health Knowledge Questionnaire (MHKQ) [33]. It is a three-factor self-reported scale consisting of 14 items. However, in this study, we only used 10 questions from two of the scale’s dimensions: knowledge of the characteristics of mental health and mental disorders; and belief in the epidemiology of mental disorders. Responses were rated on a 5-point Likert scale (1 = strongly disagree; 5 = strongly agree). Items 7–10 are reverse scored. Higher scores correspond to higher levels of mental health literacy. The European Portuguese version of the MHKQ presents a very good internal consistency (Cronbach’s alpha) (0.85 for the total scale). Job satisfaction was assessed using the European Portuguese version of the Warr et al. scale [34]. It is a self-report scale consisting of 15 items. Respondents rate items on a 7-point Likert scale ranging from 1 (very unsatisfied) to 7 (very satisfied). Total scores are calculated by summing all 15 items. Higher scores indicate higher job satisfaction. The European Portuguese version of the scale presents a very good internal consistency (Cronbach’s alpha) (0.93 for the total scale).

The potential vulnerability factors were assessed by measuring the following variables: psychological vulnerability and the number of working hours per week. The psychological vulnerability was assessed by the European Portuguese version of the Psychological Vulnerability Scale (PVS) [35]. The PVS is a six-item scale, and each item response is rated from 1 (does not describe me at all) to 5 (describes me very well). Possible total scores ranged from 6 to 30, with higher scores indicating greater psychological vulnerability. The European Portuguese version of the PVS presents good internal consistency (Cronbach’s alpha) (0.73). Finally, the number of working hours per week was assessed by asking a single open-ended question: “Usually, how many hours do you work per week?”.

### 2.5. Statistical Analysis

IBM SPSS version 25 (IBM, Armonk, NY, USA) was used for data analysis. Descriptive characteristics of the sample were obtained using absolute and relative frequencies (categorical variables) or mean and standard deviation (SD) (for quantitative variables). Multivariate analyses using multiple linear regressions (using the enter method) were performed to identify which potential protective and vulnerability factors influenced municipal workers’ depressive symptoms, anxiety, stress, and burnout. The significance level was set at 0.05.

### 2.6. Ethical Considerations

This study followed the Declaration of Helsinki and was approved by the Ethics Committee of the Portuguese Society of Mental Health Nursing (protocol code 02/LS/2021, 30 June 2021).

## 3. Results

### 3.1. Characteristics of Study Participants

A total of 115 municipal workers agreed to participate in the study and completed the data collection tool (response rate = 33.82%). There were no missing answers (item nonresponse). The participants’ mean age was 46.23 years (SD = 9.35), the mean number of years in the current job was 16.23 years (SD = 11.02), and the mean number of working hours per week was 32.33 (SD = 9.34). Table 1 summarises other demographic and clinical characteristics of the sample.

### 3.2. Outcome Data

The municipal workers’ mental health was measured by examining their depression, anxiety, stress, and burnout levels. According to the DASS-21, the following mean scores were obtained: depression 3.16 (SD = 4.41); anxiety 2.26 (SD = 3.88); stress 3.77 (SD = 4.33). Additionally, using the OLBI, the mean score obtained for burnout was 2.64 (SD = 0.76).

The potential protective factors of municipal workers’ mental health were measured through their level of resilience, mental health literacy, and job satisfaction. The following mean scores were obtained by the different measurement tools: the CD-RISC-10—mean score of resilience 28.28 (SD = 5.71); the MHKQ—mean score of mental health literacy 40.08 (SD = 4.42); and the Warr et al. scale—mean score of job satisfaction 61.25 (SD = 15.32). 

Finally, the potential vulnerability factors of municipal workers’ mental health were measured by their level of psychological vulnerability and the number of working hours per week. Using the PVS, the municipal workers’ psychological vulnerability mean score was 14.07 (SD = 5.51), and the single open-ended question showed the mean number of working hours per week, 32.33 (SD = 9.34).

### 3.3. Main Results

#### 3.3.1. Influence of the Potential Protective and Vulnerability Factors on Municipal Workers’ Depressive Symptoms

Multiple linear regression was used to test if resilience, mental health literacy, job satisfaction, psychological vulnerability, and the number of working hours per week had a significant influence on municipal workers’ depressive symptoms. The overall regression was statistically significant (*R*^2^ = 0.44, *F*(5, 109) = 17.27, *p* < 0.01). Detailed information is presented in Table 2.

#### 3.3.2. Influence of the Potential Protective and Vulnerability Factors on Municipal Workers’ Anxiety Level

Multiple linear regression was used to test if resilience, mental health literacy, job satisfaction, psychological vulnerability, and the number of working hours per week had a significant influence on municipal workers’ anxiety levels. The overall regression was statistically significant (*R*^2^ = 0.36, *F*(5, 109) = 12.34, *p* < 0.01). Detailed information is presented in Table 3.

#### 3.3.3. Influence of the Potential Protective and Vulnerability Factors on Municipal Workers’ Stress Level

Multiple linear regression was used to test if resilience, mental health literacy, job satisfaction, psychological vulnerability, and the number of working hours per week had a significant influence on municipal workers’ stress levels. The overall regression was statistically significant (*R*^2^ = 0.35, *F*(5, 109) = 11.80, *p* < 0.01). Detailed information is presented in Table 4.

#### 3.3.4. Influence of the Potential Protective and Vulnerability Factors on Municipal Workers’ Burnout

Multiple linear regression was used to test if resilience, mental health literacy, job satisfaction, psychological vulnerability, and number of working hours per week had a significant influence on municipal workers’ burnout. The overall regression was statistically significant (*R*^2^ = 0.49, *F*(5, 109) = 21.29, *p* < 0.01). Detailed information is presented in Table 5.

## 4. Discussion

The primary aim of this study was to examine the influence of potential protective and vulnerability factors on municipal workers’ depressive symptoms, anxiety and stress levels, and burnout. A sample of 115 participants was obtained, of which 62.6% were female, with an average age of 46.23 years (SD = 9.35) and a mean number of years in the current job of 16.23 (SD = 11.02). According to the Instituto Nacional de Estatística [National Institute of Statistics] [36], in 2021, in the Municipality of Felgueiras, about 51.7% of the inhabitants were female, while 48.3% were male. Moreover, 69.3% of the population was of working-age, ranging between 15 and 64 years old, and 48.8% of the population was married. The unemployment rate in this municipality was around 5.7%, below the overall national average. Therefore, the sociodemographic characteristics of this sampling seem to match the characteristics of the general population of the municipality. However, the generalizability of the results is limited due to the lack of data in the literature about other sociodemographic data of this population. 

Through the DASS-21, the depressive symptoms, anxiety, and stress were analysed to assess the workers’ mental health. As for the depressive symptoms, the mean value found was 3.16 (SD = 4.41). This score was slightly higher than the one found in the study by Henry and Crawford [37], in the United Kingdom, with a non-clinical sample of the adult population (*n* = 1794), in which the calculated value was 2.83 (SD = 3.87). Moreover, the present score was higher than the one found in the study by Sinclair et al. [38], conducted in the United States of America with a non-clinical sample (*n* = 499), reaching a mean value of 2.85 (SD = 4.10). In this study, anxiety obtained a mean score of 2.26 (SD = 3.88), which again is higher than the score in the study by Henry and Crawford (1.88; SD = 2.95) and Sinclair et al. (2.00; SD = 3.14). Finally, the mean score obtained for stress was 3.77 (SD = 4.33), which is lower compared to the previously mentioned studies ((4.73; SD = 4.20) [20] and (4.06; SD = 3.81)), respectively [21]. Moreover, the level of the workers’ burnout measured by the OLBI scored 2.64 (SD = 0.76), very close to the 2.69 (SD = 0.73) found in the study of the Portuguese cultural adaptation of this instrument [31]. These results demonstrate the presence of depressive symptoms and anxiety, which may be partially explained by the data collection period—amidst the COVID-19 pandemic—and the increasing number of infected people being reported daily. In a study carried out in Portugal with 10,529 participants, 70.9% of whom were active workers, the levels of depressive symptoms (5.34; SD = 5.19) and anxiety (4.42; SD = 5.11) were even higher, again somewhat explained by the data collection period, March 2020, the peak of the COVID-19 pandemic [39]. Additionally, in another study carried out in Portugal, with a sample of 207 adult individuals, the results corroborate the present findings, showing depressive symptoms and anxiety with more significant levels compared to stress.

The produced results aimed to answer the first research question showed that job satisfaction is a protective factor against the onset and development of depressive symptoms, while psychological vulnerability is a vulnerability factor relating to depressive symptoms. In a study conducted by Lopes et al. [40] with a sample of 300 professionals, the results corroborate those of this present study, as job satisfaction was considered a protective factor of mental health. Moreover, a study with 1570 female workers investigating job satisfaction and its association with health status concluded that the presence of depressive symptoms was related to job satisfaction, and that the higher the job satisfaction, the lower the female workers’ perception of depressive symptoms [41]. In this study, the psychological vulnerability was statistically significant as a vulnerability factor for depressive symptoms, corroborated by the results of the study of Østergaard et al. [42].

Concerning anxiety, job satisfaction was again a protective factor, which is in line with a study by Marneras [43] conducted with 120 nurses, in which those with low or no job satisfaction were also those exhibiting the highest anxiety levels. In addition, a study conducted with teachers [44] has also produced similar results. In this present study, psychological vulnerability was also identified as a statistically significant vulnerability factor towards anxiety. Moreover, Gallagher et al. [45] referred that the perception of psychological vulnerability has a significant effect on the onset of anxiety symptoms.

The results aimed at answering the third research question showed that psychological vulnerability is a factor of vulnerability to stress. This finding is in line with the concept of psychological vulnerability recommended by Sinclair and Wallston [25], who described psychological vulnerability as a set of conditions that promote harmful reactions to stress.

Concerning the results of the last research question, job satisfaction was identified as a protective factor and psychological vulnerability as a factor of vulnerability to burnout. The results of this study addressing the relationship between job satisfaction and burnout are in line with the findings of a study by Oliveira et al. [46], in which the absence of burnout was identified as a predictor of job satisfaction. As for the relationship between psychological vulnerability and burnout, Benincasa et al. [47] also advocated that lingering chronic psychological vulnerabilities can lead to the exhaustion of personal resources, favouring the onset or prevalence of burnout.

In this study, no statistically significant relationship was found between resilience and the different components of mental health (depressive symptoms, anxiety, stress, and burnout), which is not in line with the extended body of the literature in this area, affirming that greater resilience in the workplace is associated with better mental health [48]. Despite this trend, it is important to highlight a study conducted in Brazil, with 351 adults, suggesting that resilience does not necessarily imply lower anxiety levels [49]. Furthermore, the study by Pimenta [50] who explored the predictive role of perceived resilience on anxiety and depression in Portuguese adults with chronic pain, concluded that resilience significantly influenced the depression status, but had no significant impact on mental health.

Moreover, in this study, no statistically significant relationship was found between mental health literacy and the different components of mental health (depressive symptoms, anxiety, stress, and burnout). An interesting concept is that greater literacy is likely to lead to a greater ability to identify mental health-related signs and symptoms, which may encourage people to report these problems more often. However, a study by Bahrami et al. [51] addressing the correlation between mental health literacy and the mental health of participants did not confirm this relationship. 

In this study, no statistically significant relationship was found between the number of working hours and the different components of mental health (depressive symptoms, anxiety, stress, and burnout). Although a growing body of the literature points to this relationship [27,28], we can hypothesize that job satisfaction somewhat compensates for the number of working hours and has a more significant impact on the workers’ mental health.

Finally, this study had some limitations. Firstly, the small sample and the sampling technique may have hindered the generalizability of the results. On the other hand, the sample was recruited only from one municipality in the country. Another potential limitation is related to workers being invited to participate through an email sent by the City Council. Although workers were assured about anonymity, the way they were contacted may have led them to think otherwise (this may help explain the response rate of 33.82%). It should also be noted that the data collection period (July to December 2021) likely contributed to increased bias in the produced results on the mental health assessment domains, such as anxiety, depressive symptoms, or even burnout, because of constant daily reports of infected people with COVID-19.

## 5. Conclusions

In this study, job satisfaction was highlighted as a protective factor of the workers’ mental health. On the other hand, psychological vulnerability emerged as a factor of vulnerability to the development of workers’ depressive symptoms, anxiety, stress, and burnout. In this study, no statistically significant relationship was found between resilience, mental health literacy, the number of working hours and the different components of mental health (depressive symptoms, anxiety, stress, and burnout), which is not in line with a significant body of the literature in these areas. Nevertheless, it is interesting to reflect on the possibility that greater mental health literacy may facilitate the identification of mental health-related signs and symptoms, likely encouraging people to report these problems more often. Concerning the number of working hours, we hypothesize that job satisfaction may overlap with this variable, perceived by the workers as a more important factor in their mental health.

This study provides a substantial contribution to the theory in the field of workers’ mental health as it raises the discussion on the importance of job satisfaction versus the number of working hours. It also indicates that resilience, even if coupled with several other variables, may not be enough to guarantee good mental health. Therefore, these findings may prompt further research that corroborates or refutes these findings. Moreover, these study findings also contribute to practice by inciting decision-makers to act based on the new valuable information to develop more effective mental health promotion politics in the workplace.

The study results provide an excellent opportunity to advance the understanding of the factors influencing workers’ mental health in a specific context, which may foster the better planning of targeted interventions to promote mental health in the workplace.

## Figures and Tables

**Table 1 ijerph-19-14256-t001:** Characteristics of the study participants.

		*N* ^1^	% ^2^
sex	male	43	37.4
	female	72	62.6
marital status	single	18	15.7
	married	76	66.1
	divorced	18	15.7
	widow(er)	3	2.6
educational attainment	lower secondary education	4	3.5
	secondary education	50	43.5
	bachelor	47	40.8
	master	14	12.2
diagnosed mental disorder	yes	8	7.0
	no	107	93.0

^1^ Frequency; ^2^ Valid percent.

**Table 2 ijerph-19-14256-t002:** Influence of the potential protective and vulnerability factors on municipal workers’ depressive symptoms.

	*ß*	*t*	*p*
resilience	−0.02	0.20	0.84
mental health literacy	0.13	1.69	0.10
job satisfaction	−0.30	−3.84	**<0.01**
psychological vulnerability	0.45	5.26	**<0.01**
number of working hours per week	0.09	1.26	0.21

The bold was used to highlight the significant results.

**Table 3 ijerph-19-14256-t003:** Influence of the potential protective and vulnerability factors on municipal workers’ anxiety levels.

	*ß*	*t*	*p*
resilience	−0.01	0.09	0.93
mental health literacy	0.06	0.69	0.49
job satisfaction	−0.23	−2.85	**<0.01**
psychological vulnerability	0.46	5.04	**<0.01**
number of working hours per week	0.05	0.63	0.53

The bold was used to highlight the significant results.

**Table 4 ijerph-19-14256-t004:** Influence of the potential protective and vulnerability factors on municipal workers’ stress levels.

	*ß*	*t*	*p*
resilience	−0.05	−0.64	0.52
mental health literacy	0.10	1.19	0.24
job satisfaction	−0.14	−1.70	0.09
psychological vulnerability	0.47	5.09	**<0.01**
number of working hours per week	0.04	0.50	0.62

The bold was used to highlight the significant results.

**Table 5 ijerph-19-14256-t005:** Influence of the potential protective and vulnerability factors on municipal workers’ burnout.

	*ß*	*t*	*p*
resilience	−0.14	−1.87	0.07
mental health literacy	0.01	0.14	0.89
job satisfaction	−0.43	−5.92	**<0.01**
psychological vulnerability	0.36	4.44	**<0.01**
number of working hours per week	0.03	0.45	0.66

The bold was used to highlight the significant results.

## Data Availability

Not applicable.

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
