# Peer review of "Protective and Vulnerability Factors of Municipal Workers’ Mental Health: A Cross-Sectional Study"

_ijerph, 2022, doi:10.3390/ijerph192114256_

Round 1

Reviewer 1 Report

Dear authors

Many thanks for submitting this paper for review by myself. I found this an interesting and timely article given the circumstances in which we currently live in and what we have been through over the past 2 and a half years. I have some recommended changes which will enhance the paper further. Please see below

Abstract

Line 14, add an between to and individuals

Line 15 remove the between addressing and workers

Line 20, add of between comprised and 115

Line 26 remove the the between influence and worker and remove 's from worker's

Keywords

Too many key words. Please give the five most pertinent and if of equal value, place them in alphabetical order. 

Introduction

At end of paragraph one you cite a paper that discusses the number of individuals with a mental and substance misuse challenge in 2017, please only use data pertinent to mental health only and leave out any mentioning substance misuse as this is not the focus of the paper. 

Line 71, sentence beginning with despite existing ... please add another citation or two to strenghten arguement. 

Study design 

You mention STROBE. Please explain this guidance a little for those who do not know or are unaware of same. 

Statistical Analysis and elsewhere

You seem to put a comma where there should be a full stop, especially at SD's. Please revisit and amend as appropriate. 

Ethical Considerations

Please either have only one sentence stating that ethics was granted or go into detail re:ethical concerns for this study. Please do not do half and half as this potentially opens up queries about the ethical considerations for this study. 

Main Results

All the titles from 3.3.1 - 3.3.5 need to be in italics to differentiate them from the rest of the text. 

Discussion

Within the last paragraph of the discussion, you mentioned that there was a breach in data as the County Council emailed participants despite the supposed anonimity. Please tell us in detail how this was rectified - was a data protection officer consulted? what was there recommendation? 

Thank you once again for submitting this study for peer review. 

Author Response

We would like to thank you very much for your so relevant recommendations. We are sure they substantially helped improve the overall quality of the paper.  

Reviewer 2 Report

The concepts you ultimately want to know and analyze are not revealed, just a list. The introduction lists concepts not covered in the paper. Review the literature around the concept you are conducting your research on.

Analyze working hours by moving to the characteristics of the subject.

You wrote like this. “Our study aimed to assess the protective and vulnerability factors of municipal 112 workers’ mental health. Thus, firstly, to characterize the sample, some demographic and 113 clinical data were collected, such as sex, age, marital status, educational attainment, 114 years at the current job, and having a diagnosed mental disorder.” However, age, years of the current job are not found in table 1. Please put these in table1.

There are many concepts covered in research. You are also studying the relationship between 4 concepts (depressive symptoms, anxiety, stress, burnout) and 4 other variables. What is the theoretical background from which these variables were derived? Do not list the related literature but explain the theoretical background or framework of the concept derived from this study.

You wrote like this. “The potential protective factors of municipal workers’ mental health were measured considering their level of resilience, mental health literacy, and job satisfaction.” What is your rational? Or theoretical background? Where this thinking from?

“What is the association between municipal workers’ resilience and their mental health status?” In these research questions, do not analyze only correlation alone, but perform multiple regression analysis to check which four concepts you are curious about are more relevant.

Author Response

(The authors gave the same response as above.)

Reviewer 3 Report

Reviewer’s comments

While I find the general topic of mental health to be interesting and relevant, the reviewer thinks the manuscript could be improved. The reviewer thinks that the findings are disappointing, after the time and effort the authors have put into this manuscript. Unfortunately, the reviewer doesn’t believe that there is a strong enough theoretical contribution from this study to be considered a stand-alone research article. The study must make a significant stand-alone contribution that fundamentally changes our thinking about a topic and provides clear paths for new research. This manuscript is primarily an article that could be improved with new study aims and further analysis of the data.

The study is not clearly stated in the introduction. The aim stated in the abstract and the discussion is slightly different. Abstract: This study aimed to examine the associations between municipal workers’ resilience, mental health literacy, job satisfaction, psychological vulnerability, number of working hours per week, and mental health status.

Discussion: The primary aim of this study was to assess the protective and vulnerable mental health factors of municipal workers in Felgueiras.

Furthermore, the reviewer believes that the authors could have gone beyond simply investigating associations between variables to try to understand the paths and inter-relationships of the variables to mental health status.

In what way does the study follow the guidelines for STrengthening the Reporting of OBservational studies in Epidemiology (STROBE)?

Could you please say something about sampling, sample size, or missing responses in relation to this statement? A total of 115 municipal workers accepted to participate in the study and fulfilled the data collection tool.

The authors tell us what instruments they used in measuring constructs, but don’t tell readers how these constructs are defined.

From the average score provided, what does the outcome data mean high or low depression, anxiety, or stress in the sample? Depression 3,16 (SD=4,41); anxiety 2,26 (SD=3,88); stress 3,77 (SD=4,33). Additionally, using the OLBI, the mean score obtained for burnout was 2,64 (SD=0,76).

It is quite strange that the results showed no correlation between municipal workers’ number of working hours per week and their stress, and burnout.

The authors do not need to repeat the results in the discussion. ‘’A sample of 115 participants was obtained, being 62,6% female, with an average age of 46,23 years (SD=9,35) with a mean 253 number of years in the current job of 16,23 (SD=11,02).’’

I'd like to thank you again for considering me to review this article. I hope that my comments have helped to improve it for future submissions.

Author Response

(The authors gave the same response as above.)

Reviewer 4 Report

The reviewed paper concerned an important topic. The authors identified a correlation between some individual and organisational factors and mental health. 

Unfortunately, the manner of writing does not meet the requirements of a scientific paper. In the beginning, the authors must rise to the theory and clearly characterise the purpose of the article in this context. Only in this way will it be possible to indicate the importance of research for a better understanding of the relationship between professional activity and mental health. It must be explained why so different kind of predictors like mental health literacy, job satisfaction, psychological vulnerability, resilience and number of working hours was chosen to analyse.

The sample is small. We don't know if the sample is representative because we don't know anything about the characteristics of the population.  The collected data were analyzed only with the use of correlation coefficients, which is an insufficiently advanced method. 

The paper should also discuss the specific characteristic of a study population and how they could affect obtained results. After identifying the theoretical background more relevant literature is needed.

Author Response

(The authors gave the same response as above.)

Round 2

Reviewer 1 Report

Dear authors

Many thanks for submitting a revision to this manuscript for peer review. I hope you will find my comments encouraging and that they may serve to enhance the paper. 

Keywords - if of equal importance, please put in alphabetical order. 

Introduction - Pg.2 last paragraph. I am sure there are better references that can be added here to describe each of the emotions/conditions you seek to measure. Also on this paragraph please put in a page number as the text preceeding it is a quote. 

Pg. 3 - All paragraphs in red. I am concerned with the amount of quotation here. Please paraphrase so that there is a difference from what the source says and your interpretation of same. Also reference 18 needs a page number beside it. 

Measures - Line 200, keep the ed beside self-report. 

Statistical analysis - you mention qualitative variables, what are they? How can you have qualitative variables in a quantitative study? 

Line 227, you have (enter method)... I assume this is an error, please remove and if not please put some context behind it as it sticks out as it stands

Results - Line 244, replace accepted with agreed.

Outcome Data - Line 255 -  add examining between by and their 

Line 261 -Add through between measured and their.

Discussion - Line 341, 342 - add the following after obtain and before 62.6% - of which.

Also after 62.6% add were. 

On Pg.12 final paragraph, you mention an ethical concern but do not inform us on how you dealth with same. Please revise to include such detail. 

Conclusion - Line 606 - replace these with the

Thanks you once again for resubmitting and I look forward to reading the next revision. 

Author Response

We would like to thank you very much for your so relevant recommendations. We are sure they substantially helped improve the overall quality of the paper.  

All the suggestions were accepted, and they were marked up using the “Track Changes” function. Thank you very much for your constructive recommendations.

Reviewer 2 Report

You revised you work as I recommended. 

Author Response

(The authors gave the same response as above.)

Reviewer 4 Report

The authors significantly improved the quality of the article. 

Still, in the introduction, the research gap is not sufficiently described. 

In conclusion, it should be indicated what contribution to theory and practice are made by the research . 

Author Response

(The authors gave the same response as above.)
